# Suprapubic Cholecystectomy Improves Cosmetic Outcome Compared to Classic Cholecystectomy

**DOI:** 10.3390/jcm11154579

**Published:** 2022-08-05

**Authors:** Anas Taha, Stephanie Taha-Mehlitz, Ulrich Sternkopf, Elena Sorba, Bassey Enodien, Stephan Vorburger

**Affiliations:** 1Department of Visceral Surgery, Emmental Teaching Hospital, 3400 Burgdorf, Switzerland; 2Clarunis, Department of Visceral Surgery, University Center for Gastrointestinal and Liver Diseases, St. Clara Hospital and University Hospital, 4002 Basel, Switzerland; 3Faculty of Medicine, University of Berne, 3012 Berne, Switzerland; 4Faculty of Medicine, University of Zurich, 8006 Zurich, Switzerland

**Keywords:** suprapubic approach, cholecystectomy, laparoscopy, minimally invasive surgery

## Abstract

Currently, cholecystectomy is performed laparoscopically. While the conventional approach (CA) with four access ports persists, other methods seek to reduce trauma or to optimize cosmetic results. In this study, the safety and cosmetic outcome of a suprapubic approach (SA) were evaluated. Between 2015 and 2016, patients undergoing elective cholecystectomy either by CA or by a suprapubic approach (SA) at our institution were included. The cosmetic outcome, postoperative morbidity, operative time and length of stay were evaluated. Pictures of the site of intervention were taken 6–12 months postoperatively and rated on a scale from 1 (unsatisfying aesthetic result) to 5 (excellent aesthetic result). Five “non-medical” and five “medical” raters as well as one board-certified plastic surgeon performed the ratings. A total of 70 patients were included (*n* = 28 SA, *n* = 42 CA). The two groups did not differ in baseline characteristics (age, gender, BMI). The SA group showed a significantly better aesthetic outcome compared to the CA group 4.8 (4.8–4.9) vs. 4.2 (3.8–4.4), (*p* > 0.001). Medical raters: 4.0 (3.8–4.2) vs. 4.8 (4.6–5.0), (*p* < 0.001); non-medical raters: 4.2 (3.8–4.6) vs. 5.0 (4.8–5.0), (*p* < 0.001); plastic surgeon: 4.0 (4.0–4.0) vs. 5.0 (5.0–5.0), (*p* < 0.001). Fair interrater consistency was demonstrated with an ICC of 0.47 (95% CI = 0.38–0.57). No significant difference in the complication rate (1 (3.5%) in SA vs. 6 (14%) in CA, (*p* = 0.3)), or the operating time 66 (50–86) vs. 70 (65–82) min, (*p* = 0.3), were observed. Patients stayed for a median of three (3–3) days in the SA group and 3 (3–4) days in the CA group (*p* = 0.08). This study demonstrated that the suprapubic approach is an appropriate alternative to conventional laparoscopic cholecystectomy, presenting a better cosmetic outcome with a similar complication rate.

## 1. Introduction

Laparoscopic cholecystectomy (LC) is one of the most common surgical procedures performed by general surgeons worldwide. In the USA alone, between 750,000 and 1,000,000 cholecystectomies are performed each year [1]. In 1992, only seven years after its introduction in Germany, the laparoscopic approach (CA) became the gold standard for cholecystectomy in patients with symptomatic cholecystolithiasis [1,2,3,4]. Since then, laparoscopic cholecystectomy has further evolved to become as minimally invasive as possible, with smaller incisions as well as a reduced number of trocars needed [5].

In recent years, novel minimally invasive approaches have been tested. Techniques such as natural orifice transluminal endoscopic surgery (NOTES) and single-incision laparoscopic cholecystectomy (SILC) offered good cosmetic results and faster patient recovery [6,7,8]. However, their disadvantages, like a flat learning curve, the requirement of expensive access ports and specialized instruments, paired with higher intra- and postoperative morbidity, hindered their widespread application. Namely, the higher risk of herniation at the access port in SILC [8,9,10] trumped their potential advantage. Moreover, no significant benefits in patient satisfaction have been shown when comparing SILC with classic four-port LC [11].

The suprapubic approach (SA) presents a simpler and cheaper alternative to the above-mentioned operating techniques. It is basically the same operation as a classic laparoscopic cholecystectomy (CA), with the difference being that the gallbladder is removed via a port that is placed near the pubic hairline (suprapubically). This port replaces the traditional access in the right hemiabdomen and avoids the subsequent lengthening of the umbilical incision for the removal of the gallbladder. This allows for less postoperative pain and improved healing of the scar. As the umbilical incision lies within the umbilical rim and the suprapubic trocar is placed at the hairline, SA results in barely visible scars [10,12].

In this retrospective matched case study, we aim to evaluate the cosmetic outcome and morbidity of the suprapubic approach.

## 2. Material and Methods

### 2.1. Patients

Patients who are older than 18 years with an American Society of Anesthesiologists (ASA) class of 1–3 that underwent elective cholecystectomy were included. Only experienced visceral staff surgeons performed the intervention. Exclusion criteria were previous abdominal surgery, ASA > 3, pregnancy or refusal to participate in the study. In this retrospective analysis, 70 patients undergoing laparoscopic cholecystectomy between January 2015 and December 2016 at the surgical department of the Teaching Hospital Emmental, Burgdorf, were included. The two groups did not differ in baseline characteristics (age, gender, BMI). Patients suitable for analysis were contacted and asked to participate in the study by written consent. The project was approved by the Swiss ethical committee Berne (2017-00384).

### 2.2. Conventional Approach (CA)

In conventional laparoscopic cholecystectomy (CA), four trocars are used. The surgeon is situated on the left side of the patient in the American position and the camera assistant stands on the same side distally. Accordingly, the 10 mm trocar for camera position was placed at the umbilicus in open technique (Hasson). Two 5 mm trocars were placed below the xiphoid and in the lower right abdomen. The 12 mm trocar was positioned in the left upper abdomen (Figure 1a). The gallbladder was extracted via the enlarged umbilical access. An intracutaneous skin closure was performed with resorbable sutures.

### 2.3. Suprapubic Approach (SA)

In the suprapubic approach (SA), the surgeon and the assistant are again on the left side in the American position. The pneumoperitoneum is established starting with the camera trocar (10 mm) at the umbilicus but in the umbilical fold in open technique (Hasson). The subxyphoidal trocar (5 mm) is placed in the same position as mentioned in the conventional method. The suprapubic approach held the large 12 mm trocar in the middle of the suprapubic hairline. The 5 mm trocar was placed in the left upper abdomen. After retracting the gallbladder fundus through the subxyphoidal port, a long grasper was inserted using the suprapubic approach to triangulate and open the angle of the cystic duct. The 5 mm port in the left upper abdomen was engaged for the dissection with a monopolar hook. The gallbladder was removed through the 12 mm suprapubic port, which was enlarged if necessary (Figure 1b). An intracutaneous skin closure was performed with resorbable sutures.

### 2.4. Outcome Measures

The first follow-up was performed six weeks after the operation, when the medical history was taken and the abdomen was examined. A second follow-up with clinical examination and photo documentation took place 6 to 12 months postoperatively. These pictures were later rated for the study. Three photographs were taken—one frontal view and one each from a 45° left and right angle, respectively. A standard setting in the same room was defined that allowed consistent exposures in order to assess the cicatrices at comparable lighting.

The pictures were rated on an ordinal scale from 1 (unsatisfying aesthetic result) to 5 (excellent aesthetic result). Scoring was performed by five “non-medical” examiners (physician assistants (PAs) in the surgical department) and by five board-certified general surgeons not directly involved in the study from other hospitals (“medical raters”). In addition, a board-certified plastic surgeon scored the pictures. The aesthetic outcome was the primary endpoint of this study.

Secondary endpoints included intra- and postoperative complications (Clavien-Dindo classification of surgical morbidity), operating time and length of hospital stay. An independent study nurse evaluated all patients for postoperative infections according to SwissNoso 30 days postoperatively.

### 2.5. Statistical Analysis

Continuous variables are reported as medians and interquartile ranges (IQRs), and categorical variables as numbers and percentages. Aesthetic scores were averaged among the five medical and five nonmedical raters, respectively. Intergroup comparisons were carried out nonparametrically using the Wilcoxon rank-sum test exact version, which is based on the “shift” algorithm described by Streitberg and Rhömel [13]. Additionally, we assessed the interrater agreement on the scores using the intraclass correlation coefficient (ICC) for consistency in a two-way model [14]. All the analyses were carried out in R version 3.6.3 (The R Foundation for Statistical Computing, 2020) [15]. A *p*-value ≤ 0.05 on a two-sided test was considered statistically significant.

## 3. Results

### 3.1. Cohort

We included a total of 70 patients (28 SA, 42 CA) in the analysis. There were no missing data. Detailed baseline characteristics and outcomes are provided in Table 1. Age, gender, BMI, and comorbidities were homogeneously distributed among groups. One complication (3.5%) was observed in the suprapubic group (acute renal insufficiency [*n* = 1]), while 6 complications (14%) occurred in the conventional group (intractable pain requiring opioids for longer than >24 h postoperatively [*n* = 3], subcutaneous hematoma, requiring clinical follow-up [*n* = 2], pancreatitis requiring endoscopic retrograde cholangiopancreatography [*n* = 1]), Table 2. Patients stayed for a median three days in the SA group and three days in the CA group (*p* = 0.08). There was no statistically significant difference in surgical operative time among the groups (66 min (50 to 86) vs. 70 min (65 to 82), *p* = 0.3).

### 3.2. Aesthetic Outcome

Figure 1c,d illustrate an exemplary picture of CA and SA six months postoperatively. The distribution of the aesthetic outcome scoring amidst raters and in-between groups is given in Figure 2. Overall, SA has a better aesthetic outcome compared to CA (4.2 (3.8 to 4.4) vs. 4.8 (4.8 to 4.9), *p* < 0.001). Both medical (4.0 (3.8 to 4.2) vs. 4.8 (4.6 to 5.0), *p* < 0.001) as well as non-medical staff (4.2 (3.8 to 4.6) vs. 5.0 (4.8 to 5.0), *p* < 0.001) rated the suprapubic approach as aesthetically superior. Similarly, the plastic surgeon gave higher aesthetic scores to the suprapubic approach (4.0 (4.0 to 4.0) vs. 5.0 (5.0 to 5.0), *p* < 0.001).

### 3.3. Interrater Agreement

Table 3 displays the analysis of the interrater agreement. Overall, the agreement for consistency was fair among all ten raters with an ICC of 0.47 (95% CI = 0.38 to 0.57). Similar interrater scores were calculated among medical staff (ICC = 0.47 (95% CI = 0.36 to 0.59)) and nonmedical staff (ICC = 0.46 (95% CI = 0.35 to 0.57)).

## 4. Discussion

Whilst cholecystectomies became safer and less invasive over the past few decades, several surgical techniques aimed to improve cosmetic results while preserving or improving clinical outcomes [6,7,8,11,16,17]. The suprapubic approach is not a new technique, as it was already reported in the literature by Degano et al. in 1995 in Italy [18]. This fact may be connected to the widespread interest in other minimally invasive procedures like SILC and NOTES [6,7,8]. Initially, these techniques showed great promise, but there are still some shortcomings. However, SILC requires a longer operating time, has higher pain scores and an increased rate of complications, such as trocar hernias and wound-related complications [11,19,20]. Additionally, Ma et al. showed that SILC did not achieve significant improvements in overall patient satisfaction [11]. It has also been shown that additional training is needed for LC-experienced surgeons to perform the SILC operation safely [21]. In a recent review conducted by Yang et al., the NOTES technique showed excellent cosmetic results, a shorter recovery time, less postoperative pain and fewer intra- and postoperative complications [22]. On the other hand, significantly increased operating time related to a flat learning curve, even for advanced laparoscopic surgeons. Furthermore, the requirement for specialized instruments is why it makes this technique expensive [22].

In this study, we compared the suprapubic approach to CA for its cosmetic outcome. Previously, the technique of SA was reported by several institutions [10,12,23,24]. They described that cosmetic outcome, measured by the subjective parameter of patient satisfaction, improved with this approach. This study specifically sought to investigate the cosmetic outcome in a standardized setting, not by the patient or directly involved practitioner, but instead by persons independent from the treatment process. Patients, as well as independent third-party raters, unanimously found that SA yielded a significantly better cosmetic result than CA. Not surprisingly, the necessary lengthening of the umbilical incision to extract the gallbladder was the main reason why CA rated lower than SA. The placement of the incisions below the pubic hairline made the scar of the SA barely visible [23]. A study by Rabbany et al. compared the cosmetic outcome of 112 patients with suprapubic and other incisions for specimen extraction in colon surgery. The suprapubic incision was rated significantly better by the patients themselves in terms of cosmesis [25]. We assume that higher satisfaction in these patients may be caused due to less visibility of the suprapubic scar, as patients are able to cover the scar with clothing.

SA has also been found to be cheaper, to have shorter operating time and a steeper learning curve than either SILC or NOTES. The option to use conventional CA instruments makes the procedure easy to implement and cost-efficient [12]. In our study, SA and CA had similar surgical outcomes. This correlates with other reports. In addition, several recent studies showed that SA leads to less postoperative pain compared to CA [10,11,23,26,27].

We did not observe a difference in complication rate in our cohort for the two techniques in the short-term follow-up. Likewise, Griffith et al. published a study comparing 374 women with a mini-laparotomy for tissue extraction after laparoscopy, either umbilically or suprapubically. There were no significant differences in pain and infection rate at the incision site in this study. The incisional hernia rate was slightly higher at the umbilical incision site, but the difference was not statistically significant [28]. Since suprapubic incisional hernias are rare, there is little evidence of standardized suprapubic hernia repair in the literature [29]. Furthermore, the surgical treatment of a suprapubic incisional hernia is challenging due to the lack of a posterior rectus sheath and the hernia’s proximity to the bone, neural structures, vessels, and urinary tract, which might be seen as a caveat of the suprapubic trocar placement.

The mean surgical operating time in this analysis is a matter of discussion (66 min in CA vs. 70 min in SA). A large retrospective of the study the Professional Association of German Surgeons analyzed data of the digimed database during a three years course including 35,754 cholecystectomies from 86 hospitals [30]. The hospital with the lowest number of cases contributed 31 procedures, while the department with the highest number of cases contributed 913. There were 30,965 laparoscopic cholecystectomies without conversions or bile duct revisions identified. Hospitals were categorized as primary care hospitals, specialized hospitals, and university hospitals. The average operation time was 59 min for primary care centers, 66 min and 89 min for specialized and university hospitals [30]. Thus, our operating time is within the presented range. Yet, there are specialized centers with significantly shorter operating times. Voyles et al. reported an average operating time of 29 min for 99 patients [31]. The individual benchmark of operating times has to be adapted to the respective country, infrastructure, and socio-economic conditions.

## 5. Limitations

We report results from a single-center study with a limited pool of participants. Hence, all limitations of such a study design apply. The relatively low number of patients included is based on the fact that most elective cholecystectomies were not performed by experienced surgeons themselves but rather by surgeons at the beginning of their careers (mainly in a teaching situation). Hence, the vast majority of our patients did not qualify for this study.

A lower “baseline” aesthetic evaluation of a patient could influence the postoperative results. Since each patient was offered both surgical techniques, a selection bias is minimized, and the surgeon’s baseline aesthetic impression is unlikely to have influenced the outcome. Given that both groups had comparable BMI, gender and age, an unconscious baseline aesthetic impression or an already lower aesthetic evaluation preoperatively is expected to be balanced. Furthermore, using standardized photographs instead of direct observation supports the rating process, and unconscious decisions by an unintentional inclusion of the outer appearance of a patient are less likely. Additionally, to minimize an unconscious rating bias, only raters, which were not directly involved in the patient’s treatment process, evaluated the cosmetic results.

Another limitation is the number of surgeons involved in this study. Although operations performed by a single surgeon may have led to more comparable aesthetic results within a study population, it is more likely that several surgeons performed this procedure in one hospital in clinical practice. In addition, the number of participating surgeons may help that the results are more comparable to those of other hospitals.

Even though complications in the SA group were less than in the CA group, the study lacked the power to unmask a potential advantage of SA over CA, as did previous studies. Due to the short follow-up period, differences in long-term satisfaction and complication rates could not be determined. Chatzimavroudis et al. [32] described in their retrospective analysis of 1172 patients that only 0.6% of patients presented with a trocar site hernia during the 1-year follow-up, while this rate rose slightly to 0.94% after a mean follow-up of 65.86 months. This observation shows that complications like a trocar site hernia might not have occurred within our chosen timeframe of follow-up and, therefore, could be missed in our analysis.

A further limitation is that we did not use the mini-laparoscopy with a 5 mm optic, potentially improving the cosmetic outcome. In our hospital, a trocar set for cholecystectomy was introduced to optimize costs and processes, containing one 12 mm, one 10 mm and two 5 mm single-use trocars. Accordingly, we had to adapt our technique and use the trocars provided in both surgical procedures. The cosmetic outcome of laparoscopic cholecystectomies with 5 mm optics has been investigated several times. The results of the studies vary; a recently published meta-analysis by Coletta et al. could not determine any advantage for the mini-laparoscopic technique compared to the conventional approach [33]. Furthermore, the change of the umbilical incision within the umbilical fold in the SA technique can be seen as a selection bias, since this is an additional adjustment to the CA technique. For future evaluations we should use this rather hidden access for both techniques despite the potential widening for specimen salvage in the CA technique. For future evaluations we should use this rather hidden access for both techniques despite the potential widening for specimen salvage in the CA technique. Another future adaption to our current approach would be to use only 5 mm ports with the only flaw if the specimen salvage requires widening of one port site, the benefit of a 5 mm port access would be nullified [34,35].

Moreover, photographic pictures of the surgical site were used to rate the cosmetic outcome, which helped to improve the overall uniformity, but, on the other hand, could mask potential aesthetic shortcomings. We used a 5-grade Likert scale to assess the cosmetic result. This method was already used in previous studies evaluating the cosmetical outcome after procedures by plastic surgeons [36,37], but also after cholecystectomies [38]. By using a similar tool to measure the cosmetical outcome, we aimed to build a basis for comparison with other studies, which is, unfortunately, measured differently within several studies and therefore drawing comparisons is impeded. Nevertheless, the Likert scale is susceptible to a central tendency and a social desirability bias (the patient gives answers they expect the physician wants to hear). Therefore, we involved medical and non-medical raters and evaluated the rater reliability by assessing the interrater agreement using the interclass correlation coefficient.

## 6. Conclusions

This analysis of the outcome after suprapubic cholecystectomy presents an appropriate alternative to the conventional laparoscopic cholecystectomy. In our study, SA yields better cosmetic outcomes with a similar complication rate and operative time as compared with CA. Since previous studies demonstrated improved complication rates with the suprapubic approach, it should be recognized as an advanced method of laparoscopic cholecystectomy.

## Figures and Tables

**Figure 1 jcm-11-04579-f001:**
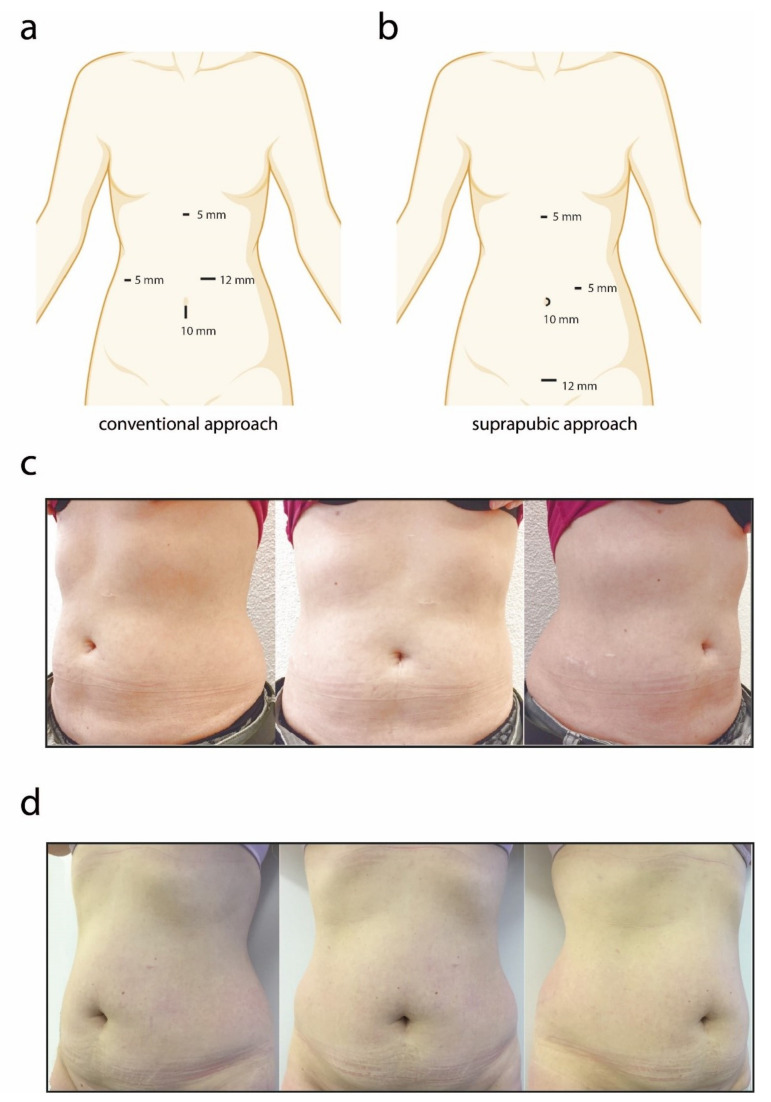
Schematic representation of the trocar positions in the abdomen in the conventional (**a**) and suprapubic (**b**) approach. Representative photographs of the SA follow-up six months postoperatively, conventional (**c**) and suprapubic approach (**d**).

**Figure 2 jcm-11-04579-f002:**
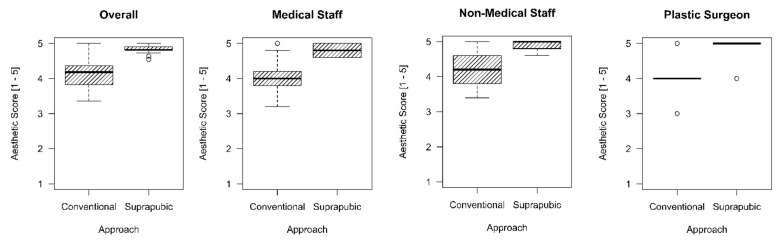
Boxplots demonstrating the distribution of the ordinal aesthetic scoring among conventional versus suprapubic laparoscopic cholecystectomy cases. The boxplots show the distribution for—from left to right—pooled overall rating, pooled medical staff rating, pooled nonmedical staff rating, and the rating from the plastic surgeon.

**Table 1 jcm-11-04579-t001:** Intergroup comparison of baseline demographics and clinical outcomes among the conventional and suprapubic laparoscopic cholecystectomy groups.

Parameter	Conventional *n* = 42	Suprapubic *n* = 28	*p*
**Demographics**			
Male, *n* (%)	6 (14%)	6 (21%)	0.650
Age, median (IQR)	53 (41 to 66)	54 (42 to 64)	0.893
BMI, median (IQR)	27 (25 to 32)	27 (25 to 32)	0.898
ASA Grade III, *n* (%)	4 (10%)	4 (14%)	0.818
**Surgical Outcome**			
Length of stay (d), median (IQR)	3 (3 to 4)	3 (3 to 3)	0.076
Surgical time (min.), median (IQR)	66 (50 to 86)	70 (65 to 82)	0.245
Complications, *n* (%)	6 (14 %)	1 (3,5%)	0.290
**Aesthetic Outcome**			
Overall (Pooled), median (IQR)	4.2 (3.8 to 4.4)	4.8 (4.8 to 4.9)	<0.001 *
Medical Staff (Pooled), median (IQR)	4.0 (3.8 to 4.2)	4.8 (4.6 to 5.0)	<0.001 *
Non-Medical Staff (Pooled), median (IQR)	4.2 (3.8 to 4.6)	5.0 (4.8 to 5.0)	<0.001 *
Plastic Surgeon, median (IQR)	4.0 (4.0 to 4.0)	5.0 (5.0 to 5.0)	<0.001 *

IQR = interquartile range; BMI = body mass index; ASA = American Society of Anesthesiologists; * *p* ≤ 0.05.

**Table 2 jcm-11-04579-t002:** Complications in Clavien-Dindo classification of surgical Outcome.

Complication (*n* = 7)	Clavien-Dindo Classification	Group
Renal insufficiency (*n* = 1)	I	Suprapubic
Intractable pain (*n* = 3)	I	Conventional
Pancreatitis (*n* = 1)	III b	Conventional
Hematoma (*n* = 2)	III b	Conventional

**Table 3 jcm-11-04579-t003:** Interrater agreement for the ordinal aesthetic rating.

Raters	No. of Raters	ICC for Consistency (95% CI)
Overall	10	0.46 (0.37 to 0.57)
Medical Staff	5	0.47 (0.36 to 0.58)
Non-Medical Staff	5	0.45 (0.34 to 0.57)

ICC = intraclass correlation coefficient; CI = confidence interval.

## Data Availability

The datasets used and/or analyzed during the current study are available from the corresponding author on reasonable request.

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
