# Peer review of "Suprapubic Cholecystectomy Improves Cosmetic Outcome Compared to Classic Cholecystectomy"

_jcm, 2022, doi:10.3390/jcm11154579_

Round 1

Reviewer 1 Report

General Remarks

This article is about the cosmetic effect of suprapubic cholecystectomy compared to classic cholecystectomy. The authors compared both procedures using rating score system with plastic surgeons and physical assistants. The methodology was interesting, however, there are several limitations to publish in this journal. Therefore, this article seems to be insufficient to convey new information to readers.

Major comments

1.      The authors described that the advantage of the SA is cosmetic outcome. However, single incision surgery or reduced port surgery are widely performed for cholecystectomy. Although SA seems to have more cosmetic effect than classic four-port surgery, the procedures which have more cosmetic effect are widely performed. The SA is not novel technique, and seems to be inferior than single-port surgery or reduced-port surgery.

2.      The authors used 12mm trocar on suprapubic area. Is 12mm trocar necessary for cholecystectomy? Numerous surgeons use only 5mm trocars except trocar for camera during laparoscopic cholecystectomy.

3.      Aesthetic outcome measurement system seems not to be objective. Although plastic surgeons and physical assistants rated the score, the method seems not to be verified.

4.      Finally, as seen from the clinical photo, it seems to that there would be no significant difference in cosmesis.

Author Response

General Remarks

This article is about the cosmetic effect of suprapubic cholecystectomy compared to classic cholecystectomy. The authors compared both procedures using rating score system with plastic surgeons and physical assistants. The methodology was interesting, however, there are several limitations to publish in this journal. Therefore, this article seems to be insufficient to convey new information to readers.

Major comments

  1. The authors described that the advantage of the SA is cosmetic outcome. However, single incision surgery or reduced port surgery are widely performed for cholecystectomy. Although SA seems to have more cosmetic effect than classic four-port surgery, the procedures which have more cosmetic effect are widely performed. The SA is not novel technique, and seems to be inferior than single-port surgery or reduced-port surgery.

We thank reviewer 1 for this critical appraisal of our work and believe that our manuscript has improved after addressing these comments.

We entirely agree that the SA technique has been described several times in the past, therefore our intention was not to introduce this technique as a recently developed technique, rather than challenging its cosmetical outcome compared to the conventional technique. We elaborated on the SILC and NOTES technique in the introduction and discussion so far. Furthermore, we described in the limitation section why we did not use a probably favourable 5-mm camera technique, still this could should be an approach for future improvements of the described technique. As noted in the limitation section, we would like to mention that the SA technique had to be performed with the same set of ports available as for the CA technique. Given the fact that we were using a trocar set for cholecystectomy that was introduced to optimize costs and processes, containing one 12 mm, one 10 mm and two 5 mm single-use trocars. Accordingly, we had to adapt our technique and use the trocars provided in both of the surgical procedures (CA and SA).

To address your major comment and explain this to the reader, we would like to kindly emphasize the following sections and add a further explanation:

Introduction

Since then, laparoscopic cholecystectomy has further evolved to become as minimally invasive as possible, with smaller incisions as well as a reduced number of trocars needed [5].

In recent years, novel minimally invasive approaches have been tested. Techniques such as natural orifice transluminal endoscopic surgery (NOTES) and single-incision laparoscopic cholecystectomy (SILC) offered good cosmetic results and faster patient recovery [6-8]. However, their disadvantages, like a flat learning curve, the requirement of expensive access ports and specialized instruments, paired with higher intra- and postoperative morbidity, hindered their wide-spread application. Namely, the higher risk of herniation at the access port in SILC [8-10] trumped their potential advantage. Moreover, no significant benefits in patient satisfaction have been shown when comparing SILC with classic four-port LC [11].

The suprapubic approach (SA) presents a simpler and cheaper alternative to the above-mentioned operating techniques. It is basically the same operation as a classic laparoscopic cholecystectomy (CA), with the difference being, that the gallbladder is removed via a port that is placed near the pubic hairline (suprapubically). This port replaces the traditional access in the right hemiabdomen and avoids the subsequent lengthening of the umbilical incision for the removal of the gallbladder. This allows for less postoperative pain and improved healing of the scar. As the umbilical incision lies within the umbilical rim and the suprapubic trocar is placed at the hairline, SA results in barely visible scars [10,12].

….

Discussion

The suprapubic approach is not a new technique, it was already reported in literature by Degano et al. in 1995 in Italy [18]. This fact may be connected to the widespread interest in other minimally invasive procedures like SILC and NOTES [6-8]. Initially, these techniques showed great promise, but there are still some shortcomings. However, SILC has longer operating time, higher pain scores and increased rate of complications, such as trocar hernias and wound-related complications [11,19,20]. Additionally, Ma et al. showed that SILC did not achieve significant improvements in overall patient satisfaction [11]. It has also been shown that additional training is needed for LC-experienced surgeons to perform the SILC operation safely [21]. In a recent review conducted by Yang et al., the NOTES technique showed excellent cosmetic results, a shorter recovery time, less postoperative pain and fewer intra- and postoperative complications [22]. On the other hand, significantly increased operating time related to a flat learning curve, even for advanced laparoscopic surgeons. Furthermore, the requirement for specialized instruments is why it makes this technique expensive [22].

SA has also been found to be cheaper, to have shorter operating time and a steeper learning curve than either SILC or NOTES. The option to use conventional CA instruments makes the procedure easy to implement and cost-efficient [12]. In our study, SA and CA had similar surgical outcomes. This correlates with other reports. In addition, several recent studies

showed that SA leads to less postoperative pain compared to CA [10,11,23,26,27].

  1. The authors used 12mm trocar on suprapubic area. Is 12mm trocar necessary for cholecystectomy? Numerous surgeons use only 5mm trocars except trocar for camera during laparoscopic cholecystectomy.

We agree the cholecystectomy is one of the most performed general surgery procedures in the world and developing techniques and instruments to improve not only the general surgical outcome, but also the cosmetical outcome is part of the development of a procedure. We aimed to explain our approach using the same trocars in SA technique as we used in the CA technique already in the reply to your first comment. Most of our patients presented with chronic cholecystitis and the extraction of the gallbladder required almost always a widening of the extraction site. Hence, the benefit of a smaller suprapubic incision would be nullified. However, as the operation can be equally done with a 5mm suprabupic incision, we consider to perform future interventions by firstly appreciating the gallbladder and then choose the size of the suprapubic trocar to further improve cosmesis. Thank you for this valuable input. We would like to address this drawback in the limitation section:

Another future adaption to our current approach would be to use only 5 mm ports with the only flaw if the specimen salvage requires widening of one port site, the benefit of a 5 mm port access would be nullified [33-34].

  1. Bender K, Lewin J, O'Rourke H, Hugh FC, O'Rourke N, Hugh TJ. Total 5-mm port approach: a feasible technique for both elective and emergency laparoscopic cholecystectomy. ANZ J Surg. 2018 Nov;88(11):E751-E755. doi: 10.1111/ans.14460. Epub 2018 Apr 24. PMID: 29687556.
  2. El-Dhuwaib Y, Hamade AM, Issa ME, Balbisi BM, Abid G, Ammori BJ. An "all 5-mm ports" selective approach to laparoscopic cholecystectomy, appendectomy, and anti-reflux surgery. Surg Laparosc Endosc Percutan Tech. 2004 Jun;14(3):141-4. doi: 10.1097/01.sle.0000129399.95866.5b. PMID: 15471020.

  1. Aesthetic outcome measurement system seems not to be objective. Although plastic surgeons and physical assistants rated the score, the method seems not to be verified.

We thank the reviewer for this important comment on the evaluation method. In our analysis we used a 5 grade Likert scale to assess the aesthetic result. This method was already used in previous studies evaluating the cosmetical outcome after procedures by plastic surgeons, but also to differentiate the aesthetic outcome after cholecystectomies. By using a similar tool to measure the cosmetical outcome we aimed to build a basis for comparison with other studies. We have to admit that the cosmetical outcome is measured differently within several studies and therefore drawing comparisons is impeded. Nevertheless, we have to agree that the Likert scale is susceptible to central tendency and to avoid a social desirability bias (the patient gives answers they expect the physician wants to hear) we involved medical and non-medical raters. Furthermore, we evaluated the rater reliability by assessing the interrater agreement using the interclass correlation coefficient, as described in the methods section. Whether the Likert scale discriminates better using 4 or 5 grades depends on the question type. But a midpoint can be a subject to satisficing behaviour, when the patient answers quickly without making up their mind building an opinion. To the best of our knowledge, a Likert scale is not validated for aesthetic results. To elaborate on this shortcoming, we would like to add the following section and literature:

We used a 5-grade Likert scale to assess the cosmetic result. This method was already used in previous studies evaluating the cosmetical outcome after procedures by plastic surgeons [36-37], but also after cholecystectomies [38]. By using a similar tool to measure the cosmetical outcome, we aimed to build a basis for comparison with other studies, which is, unfortunately, measured differently within several studies and therefore drawing comparisons is impeded. Nevertheless, the Likert scale is susceptible to central tendency and a social desirability bias (the patient gives answers they expect the physician wants to hear). Therefore, we involved medical and non-medical raters and evaluated the rater reliability by assessing the interrater agreement using the interclass correlation coefficient.

  1. Hung T, Chang W, Vlantis AC, Tong MC, van Hasselt CA. Patient satisfaction after closed reduction of nasal fractures. Arch Facial Plast Surg. 2007 Jan-Feb;9(1):40-3. doi: 10.1001/archfaci.9.1.40. PMID: 17224487.
  2. Sharma SD, Kwame I, Almeyda J. Patient aesthetic satisfaction with timing of nasal fracture manipulation. Surg Res Pract. 2014;2014:238520. doi: 10.1155/2014/238520. Epub 2014 Jan 2. PMID: 25374948; PMCID: PMC4208588.
  3. Hauters P, Auvray S, Cardin JL, Papillon M, Delaby J, Dabrowski A, Framery D, Valverde A, Rubay R, Siriser F, Malvaux P, Landenne J. Comparison between single-incision and conventional laparoscopic cholecystectomy: a prospective trial of the Club Coelio. Surg Endosc. 2013 May;27(5):1689-94. doi: 10.1007/s00464-012-2657-x. Epub 2012 Dec 7. PMID: 23224032.

  1. Finally, as seen from the clinical photo, it seems to that there would be no significant difference in cosmesis.

Thank you for raising this important comment. The difference of the cosmetical result demonstrated in the exemplary images is only marginally. We used photographs of the surgical sites to rate the cosmetic outcome, which helped to improve the overall uniformity, but, on the other hand, could mask potential aesthetic shortcomings. In both procedures the cosmetical result was quite satisfactory, as reflected in the relatively high rating for both, the CA and SA technique. Nevertheless, we aimed to outline with the chosen photographs the main difference that is visible in the photographs taken, which is the 12 mm port scar in the left upper quadrant of the patient with CA, which is (being a 5 mm port scar) less prominently visible in the SA patient and the umbilical incision. We agree, it is challenging to demonstrate the difference in these exemplary photographs of one patient, especially since the cosmetic overall result after cholecystectomy is usually quite satisfying. To face this challenge, we used the graphs in figure 2 as well to demonstrate the measured rating difference. We addressed this shortcoming briefly in our limitation section.

Moreover, photographic pictures of the surgical site were used to rate the cosmetic outcome, which helped to improve the overall uniformity, but, on the other hand, could mask potential aesthetic shortcomings.

However, overall I believe this manuscript to be an interesting article for a surgical audience describing a potential aesthetic improval in one of the most common surgeries in the world.

Reviewer 2 Report

Dear Mr. Taha, dear Mr. Vorburger, dear co-authors,

Thank You for providing your manuscript Suprapubic cholecystectomy improves cosmetic outcome com-2 pared to classic cholecystectomy", in which you describe an alternative approach in performing laparoscopic cholecystectomy with beneficial aesthetic results.

Your article was very interesting, especially for a surgical audience, well-structured and clearly readable. Your methods used are comprehensible, provided tables and figures are clear and coherent. The discussion is well-balanced, potentially upcoming questions are adressed adequately.

As a matter of fact, a retrospective single-center study comes with certain limitations, as mentioned after the discussion part within your manuscript. Therefore, the selection period (2 years) seems a bit short for a retrospective single-center study, as previous articles showed even more beneficial results in regard to complications. Were regular follow-up photographs only taken in this time period? Maybe you can briefly comment on this.

Tempted by your article, many surgeons would potentially like to apply the suprapubic technique in their clinical routine when performing cholecystectomies. In my eyes, this audience would be happy to have a slightly more thorough description of the applied method. For example: What is the position of the operating surgeon, how is the positioning of the patient? Through which trocar do you insert your "primary" instruments in order to perform the surgery? A brief description with more details would surely be appreciated.

As you describe yourself, the periumbilical incision is regarded to be mostly relevant for the postoperative aesthetic result, especially when the incision needs to be lengthened for extraction of the gallbladder (line 186-187). In the SA group, this incision is "hidden" in the umbilical fold, therefore supposedly relevantly adding to the beneficial aesthetic result of the SA group. However, this incision can also be placed in the umbilical fold when applying the conventional technique, leading to an equally satisfying (periumbilical) aesthetic result - if the incision does not need to be lengthened. In my eyes, this can be regarded as a selection bias. I would suggest that you comment on this when revising the manuscript.

However, overall I believe this manuscript to be an interesting article for a surgical audience describing a potential aesthetic improval in one of the most common surgeries in the world.

Author Response

General Remarks

This article is about the cosmetic effect of suprapubic cholecystectomy compared to classic cholecystectomy. The authors compared both procedures using rating score system with plastic surgeons and physical assistants. The methodology was interesting, however, there are several limitations to publish in this journal. Therefore, this article seems to be insufficient to convey new information to readers.

Thank You for providing your manuscript Suprapubic cholecystectomy improves cosmetic outcome compared to classic cholecystectomy", in which you describe an alternative approach in performing laparoscopic cholecystectomy with beneficial aesthetic results.

Your article was very interesting, especially for a surgical audience, well-structured and clearly readable. Your methods used are comprehensible, provided tables and figures are clear and coherent. The discussion is well-balanced, potentially upcoming questions are adressed adequately.

As a matter of fact, a retrospective single-center study comes with certain limitations, as mentioned after the discussion part within your manuscript. Therefore, the selection period (2 years) seems a bit short for a retrospective single-center study, as previous articles showed even more beneficial results in regard to complications. Were regular follow-up photographs only taken in this time period? Maybe you can briefly comment on this.

We thank the reviewer for this comment. Actually, the first assessment took place within 30 days after surgery by a study nurse, we mentioned this briefly in the manuscript. At this timepoint no other pictures were taken. Our first follow-up to detect other early complications was done at about 6 weeks post surgery. Yet, for the cosmetical result we have taken only once photographs during another follow-up that took place between 6-12 months after the procedure itself and during this follow-up, the patient was assessed again for complications. We agree that during a longterm follow-up other complications like a trocar hernia could be detected.

A brief description of the follow-up and actions taken was given in the manuscript so far. Furthermore, we are happy to comment on the missing longterm follow-up within our limitation section:

Outcome Measures

The first follow-up was performed 6 weeks after the operation, when medical history was taken and the abdomen was examined. A second follow-up with clinical examination and photo documentation took place 6 to 12 months postoperatively.

Secondary endpoints included intra- and postoperative complications (Clavien-Dindo classification of surgical morbidity), operating time and length of hospital stay. An independent study nurse evaluated all patients for postoperative infections according to SwissNoso 30 days postoperatively.

Even though complications in the SA group were less than in the CA group, the study lacked the power to unmask a potential advantage of SA over CA, as did previous studies. Due to the short follow-up period, differences in long-term satisfaction and complication rates could not be determined. Chatzimavroudis et al. [32] described in their retrospective analysis of 1,172 patients that only 0.6% of patients presented with a trocar site hernia during the 1-year follow-up. While this rate rose slightly to 0.94% after a mean follow-up of 65.86 months. This observation shows that complications like a trocar site hernia might not have occurred within our chosen timeframe of follow-up and, therefore, could be missed in our analysis.

  1. Chatzimavroudis G, Papaziogas B, Galanis I, Koutelidakis I, Atmatzidis S, Evangelatos P, Voloudakis N, Ananiadis A, Doundis A, Christoforidis E. Trocar site hernia following laparoscopic cholecystectomy: a 10-year single center experience. 2017 Dec;21(6):925-932. doi: 10.1007/s10029-017-1699-3. Epub 2017 Oct 25. PMID: 29071498.

Tempted by your article, many surgeons would potentially like to apply the suprapubic technique in their clinical routine when performing cholecystectomies. In my eyes, this audience would be happy to have a slightly more thorough description of the applied method. For example: What is the position of the operating surgeon, how is the positioning of the patient? Through which trocar do you insert your "primary" instruments in order to perform the surgery? A brief description with more details would surely be appreciated.

We thank reviewer 2 for this valuable comment. We would like to add some more information for the reader about the procedure itself. To address this request, we added the following information into the methods description:

2.2. Conventional Approach (CA)

In conventional laparoscopic cholecystectomy (CA), four trocars are used. The surgeon is situated on the left side of the patient in American position and the camera assistant stands on the same side distally. Accordingly, the 10 mm trocar for camera position was placed at the umbilicus in open technique (Hasson). Two 5 mm trocars were placed below the xiphoid and in the lower right abdomen. The 12 mm trocar was positioned in the left upper abdomen (Figure 1 a). The gallbladder was extracted via the enlarged umbilical access. Intracutaneous skin closure was performed with resorbable sutures.

2.3. Suprapubic Approach (SA)

In the suprapubic approach (SA), the surgeon and the assistant are again on the left side in American position. The pneumoperitoneum is established starting with the camera trocar (10 mm) at the umbilicus but in the umbilical fold in open technique (Hasson). The subxyphoidal trocar (5 mm) is placed in the same position as mentioned in the conventional method. The suprapubic approach held the large 12 mm trocar in the middle of the suprapubic hairline. The 5 mm trocar was placed in the left upper abdomen. After retracting the gallbladder fundus through the subxyphoidal port, a long grasper was inserted using the suprapubic approach to triangulate and open the angle of the cystic duct. The 5 mm port in the left upper abdomen was engaged for the dissection with a monopolar hook. The gallbladder was removed through the 12 mm suprapubic port, which was enlarged if necessary (Figure 1 b). Intracutaneous skin closure was performed with resorbable sutures.

As you describe yourself, the periumbilical incision is regarded to be mostly relevant for the postoperative aesthetic result, especially when the incision needs to be lengthened for extraction of the gallbladder (line 186-187). In the SA group, this incision is "hidden" in the umbilical fold, therefore supposedly relevantly adding to the beneficial aesthetic result of the SA group. However, this incision can also be placed in the umbilical fold when applying the conventional technique, leading to an equally satisfying (periumbilical) aesthetic result - if the incision does not need to be lengthened. In my eyes, this can be regarded as a selection bias. I would suggest that you comment on this when revising the manuscript.

We thank reviewer 2 for raising this important comment. It is correct that we used for the SA technique besides the suprapubic incision also an adapted umbilical incision. To address your comment, we would like to add the following addition to the limitation section:

Furthermore, the change of the umbilical incision within the umbilical fold in SA technique can be seen as a selection bias, since this is an additional adjustment to the CA technique. For future evaluations we should use this rather hidden access for both techniques despite the potential widening for specimen salvage in the CA technique. For future evaluations we should use this rather hidden access for both techniques despite the potential widening for specimen salvage in the CA technique. Another future adaption to our current approach would be to use only 5 mm ports with the only flaw that again the specimen salvage requires widening of one port site in certain cases [34,35].

However, overall I believe this manuscript to be an interesting article for a surgical audience describing a potential aesthetic improval in one of the most common surgeries in the world.

Round 2

Reviewer 1 Report

The authors faithfully responded to the reviewer's comments. However, it is judged inappropriate because there is still insufficient new information to be published in this journal.